# Input-Output Efficiency of Economic Growth: A Multielement System Perspective

## Lei Kang [1] and Zhouying Song [1,2,*]

[1]    Key Laboratory of Regional Sustainable Development Modeling, Institute of Geographic Sciences and Natural Resources Research, CAS, Beijing 100101, China; kanglei@igsnrr.ac.cn

[2]    College of Resources and Environment, University of Chinese Academy of Sciences, Beijing 100049, China

*    Correspondence: songzy@igsnrr.ac.cn

**Abstract:** Achieving sustainable and efficient economic development involves the pursuit of a model with low input, low emissions, and high yield. One approach to this is by considering input-output efficiency, which has been studied by many previous studies. However, existing literature mainly tend only to give an overall evaluation of regional input-output efficiency, which is unable to reveal the structure and components within the input-output system. This paper aims to overcome this problem by a systematic examination and measuring the resource efficiency, socio-economic efficiency, and environmental efficiency of separate subsystems using the Super-DEA model. The overall efficiency of 30 Chinese provinces from 2000 to 2015 is analyzed, along with each subsystem's efficiency. The results show: (i) The overall input-output efficiency, resource efficiency, and socio-economic efficiency of the eastern region are relatively high. The efficiency of the northeastern region has performed poorly. Although the efficiency of the central and western regions is not high, their resource efficiency and socio-economic efficiency have risen in the last decade; (ii) Environmental efficiencies are markedly lower than the levels of the other two subsystems. Most western and northeastern provinces increased in rank, while most eastern and central provinces fell. (iii) Provinces can be divided into three categories, such as resource, socio-economic, and environmental efficiency-constrained provinces. Finally, we discuss the reasons driving the spatiotemporal pattern of China's input-output efficiency and improvement policies.

**Keywords:** input-output efficiency; resource efficiency; socio-economic efficiency; environmental efficiency; constraint factor; China

## 1. Introduction

The UN 2030 Sustainable Development Goals represent the continuation and upgraded version of the Millennium Development Goals, and simultaneously contain new breakthroughs. One such breakthrough is the added emphasis on the quality and effectiveness of economic development, as exemplified by the proposition in goal eight to "promote sustained, inclusive, and sustainable economic growth". In the 21st century, the question of how to raise the quality and effectiveness of economic development while pursuing economic growth has become a topic of concern for the sustainable development of all countries [1,2]. Achieving sustainable and efficient economic development involves the pursuit of a model that features low input, low emissions, and high yield [3]. This is an important way to achieve harmony between society and the economy on the one hand and resources and the environment on the other.

One approach to this is by considering input-output efficiency, which has become a research topic of great concern in recent years. It is an important indicator for measuring resource allocation efficiency, factor input-output capacity, and sustainability of economic growth. Related studies are

mainly focused on two aspects. One involves selecting a single resource factor, such as land, water, or energy resources, and examining the efficiency of use or input-output efficiency of that particular factor. For example, Xie et al. [4] used a non-radial directional distance function approach to evaluate the green efficiency of arable land use to reveal a u-shaped trend of an initial decline followed by a later increase. Many studies focus on measuring water use efficiency, as increasing water pollution has become a serious problem worldwide, and the policy implications of bringing improvements [5–7]. At the same time, there are studies evaluating the efficiency of specific types of energy sources [8], with energy efficiency evaluation research playing an important role in improving the efficiency of energy production and utilization [9]. Obviously, the analysis of the input-output efficiency of a single type of resource can reflect the efficiency status of a certain type of resource element. However, because the focus is only on a certain level of the economic system, it may ignore the status and impact of other economic development factors, which is not conducive to the optimal allocation of input and output elements and improvement of the overall economic development quality. Therefore, many scholars are keen to measure the input-output efficiency by gathering various factors from a comprehensive perspective.

The other aspect involves selecting a specific entity of economic development, and evaluating its overall efficiency with establishing a comprehensive index system covering various input factors. Tao et al. [10], for example, employed a non-separable input/output slack-based model, abbreviated as SBM, to measure provincial green economic efficiency during 1995–2012. It showed that there are larger interregional differences in green economic efficiencies. The highest efficiency was recorded in the southern coastal region, followed by those in the eastern coastal and northern coastal regions, and the lowest efficiency in the northwestern region. Ren et al. [11] introduced undesirable output into the measurement of total factor productivity based on the Malmquist–Luenberger index model to evaluate the green efficiency of the marine economy under environmental constraints of eleven coastal regions. They found that the total factor productivity of China's marine economy without considering undesirable output is markedly higher than that considering it. Moutinho et al. [12] used an output-oriented model with two specifications (variable and constant returns to scale) to estimate the efficiency of 26 European countries, and found large disparities and a significant change in the economic and environmental efficiency trend in European countries. They suggest that the share of renewable and non-renewable energy sources is important to explain differences in emissions. Huang et al. [1] measured the urban eco-efficiency of 273 cities in China from 2003 to 2015 based on data envelopment analysis (DEA), revealing a general overall improvement due to the contribution of urban clusters. The results indicate that the improvement of the urban cluster is conducive to enhancing urban eco-efficiency. There is a "core-periphery" spatial structure in the process of urban cluster development, and the impact of urban clustering on the eco-efficiency in core cities is stronger than in the periphery cities. Yin et al. [13] used eco-efficiency as an indicator to measure urban sustainable development. They found that almost half of the cities are fairly eco-efficient. The inefficient cities are mainly located in the southwest and northwest of China, which are the undeveloped economic zones. In contrast, some of the eco-efficient cities have more environmental pollution and consume more land, energy, and water. Besides, environmental efficiency in the process of economic development has attracted attention [12,14]. For example, Chen et al. [14] constructed the input and output index system for environmental efficiency evaluation based on the panel data of provinces in China, and found that the environmental efficiency was low from 2001 to 2010. The Chinese provinces should make greater efforts to be efficient because most provinces are far away from the production frontier. Many other scholars have conducted evaluation studies on green economic efficiency [15,16], ecological efficiency [17–19], urban efficiency [20,21], resource and environmental efficiency [22] in specific areas. The studies' key words include resources, environment, and sustainability, all from the input-output perspective, with the goal of minimizing the impact of resources and the environment and maximizing economic output. These studies follow the concept that economic development cannot sacrifice resources and the environment. The ultimate goal is to measure the degree of coordination between regional economic

development, ecological consumption, and environmental protection, reveal the regional development quality, and guide regional sustainable development.

However, the extensive studies of regional input-output efficiency, although helpful in improving environmental protection and sustainable development, mostly regard regional input-output efficiency as a completely entire system. They mainly provide only an overall evaluation based on a series of input-output indicators that is unable to analyze the properties of the structure and components within the system. In fact, the economic activities involve the consumption of various factors, which means the input-output factor system involved is very complicated. In order to avoid the imbalance of the evaluation system due to excessive emphasis on certain factors, the input-output efficiency evaluation often selects a series of elements as input indicators in order to pursue the comprehensiveness of the evaluation process, and then conducts a comprehensive evaluation. This method makes it easy to obscure the effect characteristics of input factors in some aspects, so that the evaluation results would stay at the system level and cannot penetrate to the element level. In reality, there can be obvious discrepancies between the efficiency levels of subsystems obtained by dividing regional input-output efficiency according to the different input factors involved, with different subsystems having their own efficiency characteristics. Previous studies usually failed to monitor the efficiency levels of each subsystem separately from the element type level, making it challenging to identify the bottlenecks in input-output efficiency. Therefore, it is necessary to characterize the input-output efficiency by different subsystems to reflect the differences and effects of different types of elements, and then analyze the problems existing in the entire economic system. This is conducive to minimizing resource input and environmental losses in the process of economic output and maximizing economic benefits.

In this paper, we seek to fill these gaps. This study attempts to decompose complex efficiency evaluation based on systematic thinking, measuring resource efficiency, socio-economic efficiency, and environmental efficiency by dividing the entire input-output system into three subsystems. The study involves the application of the super-efficiency data envelope analysis (DEA) model, which has the advantage of being able to identify the ranks of effective decision-making units through frontier conversion.

China has experienced four decades of rapid economic development since the launch of its reform and opening-up policy in the 1980s. However, this type of development model primarily relies on high resource and environmental input and low labor costs, and has resulted in unavoidable issues of high resource consumption and severe environmental pollution as the Chinese economy has grown [23]. Between 1978 and 2016, China's total energy consumption increased from 571 million to 4.36 billion tons of standard coal [24], while solid, liquid, and gaseous industrial emissions have remained high [25]. The country has been constantly facing new problems and challenges over environmental quality and environmental protection, particularly since the 1990s, when several highly polluting and energy-intensive industries relocated to China [26]. As the 2014 Annual Report on Low-Carbon Economic Development indicated, China's economic development has the highest resource consumption and emissions of a variety of pollutants and is beginning to reach the limit of the country's environmental capacity [27]. These problems have seriously impeded the sustainable development of China's economy. The 2016 report further indicated that China's future economic development must focus on raising quality and efficiency, and that economic development must be transformed more quickly to promote a green and low-carbon shift [27]. The 19th National Congress of the Communist Party of China introduced the concept of "promoting the elevation of economic development quality, efficiency, and dynamism". Therefore, it is important to evaluate China's economic efficiency as well as its input-output situation for promoting its economic growth pattern. China's high-speed economic development is facing challenges, including huge resource consumption, and increasingly severe environmental deterioration. Under this background, transforming the economic growth mode and improving the input-output efficiency of economic development is an inevitable choice for China to achieve sustainable development. It is necessary to clarify the problems existing in the different aspects

of economic activities in order to coordinate the relationship between economic growth, resource conservation, and environmental protection to achieve sustainable development.

In this study, we analyze the overall efficiency of 30 Chinese provinces from 2000 to 2015 (the original data is in Supplementary Materials), along with each subsystem's efficiency, including resource efficiency, socio-economic efficiency, and environmental efficiency. On the one hand, the realization of the UN 2030 Sustainable Development Goals depends on the coordination between socio-economic cost, resource conservation, and environmental protection. Besides, looking through the sustainable development status in China, we can find out that the challenges of China's sustainable development are also mainly referred with high resource consumption, high social-economic costs, and serious environmental pollution. On the other hand, existing studies reveal that the input factors mainly comprise natural resources, socioeconomic, and environmental factors, which will be explained in detail in the following part. We also discuss their impact on the overall efficiency of the entire system, to provide a more complete examination of the characteristics of the country's spatiotemporal evolution of regional input-output efficiency and identification of the factors restricting increased economic efficiency. Besides, it is easier to grasp the input and output efficiency of a single subsystem. We hope the study can not only reflect China's entire economic activities through overall input-output efficiency, but also characterize the input and output efficiency of natural resource elements, socio-economic factors, and environmental factors. Furthermore, we try to find out the shortcomings that restrict the regional sustainable and efficient development more accurately, and thus provide a scientific basis for achieving green development.

The main content of this paper is arranged as follows. First, the research framework is presented, including the conceptual model and the decomposition into subsystems. Next, the methods used, case study, and data sources are described. This is followed by the results and analysis, covering spatiotemporal evolution and input-output efficiency, including both the overall system and each subsystem, together with a constraint analysis and categorization of input-output efficiency. The final section contains the conclusions and some discussion.

## 2. Conceptual Framework

Input-output efficiency, which is essentially the ratio between the yield of economic activities and the amount of resource and environmental input, is the key to sustainable economic development. Its calculation is based on a large and complex system that provides product output through the input and consumption of different factors, each of which corresponds with different subsystems at a subsidiary level to overall input-output efficiency. As there are both different degrees of correlation and relative independence between the various subsystems, there can be differences between separate subsystems and the overall system within the same region as well as in terms of input-output efficiency. However, previous studies generally obtained an overall efficiency value based on the entire input-output system and cannot reflect the input-output characteristics of its subsystems. As shown in Figure 1, this study starts with the "independence" of multiple subsystems and "dissects" the regional overall input-output efficiency system based on the input of many factors. Adopting a multidimensional perspective, a separate quantitative calculation is carried out of the regional overall input-output efficiency and its subordinate subsystems.

The input factors in existing studies mainly comprise water resources, land resources, forest resources, energy resources, investment, labor, technological innovation, intellectual property rights, and environmental capacity, and so on, as summarized in Table 1. In light of their diversity, the various factors are divided into the three categories of natural resource, socioeconomic, and environmental factors, which serve as the basis for decomposing the overall input-output efficiency systems into their separate subsystems (Figure 1, Table 2).

**Table 1.** Summary of eco-efficiency research object, methods, and indicators.

| Research Object | Method | Economic Accounting Indicators | Resources Input Indicators | Environmental Index | Authors | Publication Date |
|---|---|---|---|---|---|---|
| Area green efficiency | DEA-Malmquist model | GDP | Labor, capital, energy consumption, built-up area | Sulfur dioxide emission | Wenhua Yuan, Jianchun Li, Li Meng, et al. [15] | 2019 |
| Green economic efficiency | SBM-DEA | GDP | Labor, capital stock, energy consumption | Industrial waste gas, wastewater, solid waste | Chenfeng Zhuo, Feng Deng [16]. | 2020 |
| Eco-efficiency | Undesirable output SBM Model | GDP | Land area, expenditure on scientific undertakings, employees, total fixed asset investment | Industrial wastewater, industrial sulfur dioxide emissions, industrial smoke (powder) dust emissions | Qianqian Liu, Shaojian Wang, Bo Li, et al. [17] | 2020 |
| Urban agglomerations Eco-efficiency | super-efficient DEA | GDP | Energy consumption, power consumption, water consumption, land area, employees | Wastewater, waste gas, solid waste emissions | Lina Fu, Xiaohong Chen, Zhihua Leng [19]. | 2013 |
| Urban agglomeration input and output efficiency | Bootstrap-DEA model | GDP, industrial value added, total retail sales of consumer goods | Investment in fixed assets, land area, information element input, employees | Wastewater, waste gas, solid waste emissions | Chuanglin Fang, Xingliang Guan [21] | 2011 |
| Urban resource-environmental efficiency | DEA model | Economic indicator (GDP), ecological indicator (forest coverage rate) | Total employment, total investment in fixed assets, urban construction land area, total electricity consumption, total water supply | Intensity of industrial wastewater, emissions intensity of industrial sulfur dioxide, intensity of industrial solid waste | Xiaoping Zhang, Yuanfang Li, Wenjia Wu. [22] | 2014 |

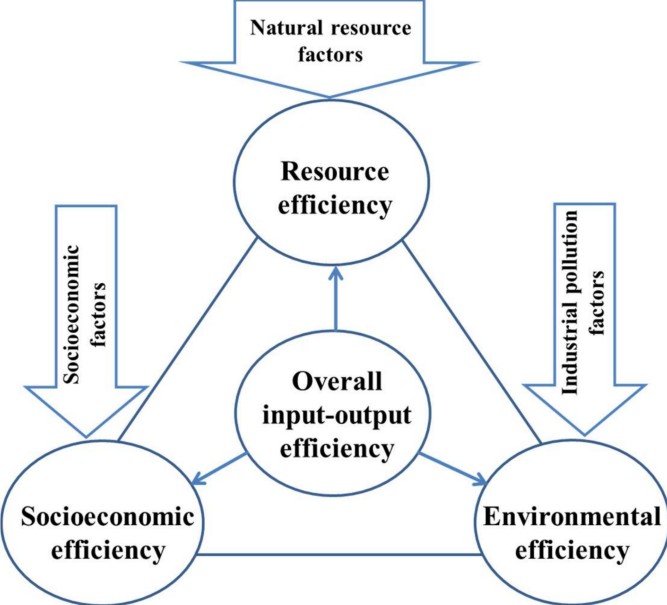

**Figure 1.** Conceptual framework.

### 2.1. Natural Resource Factors

Water, land, and energy resources represent the foundation of the continued evolution of the human-land relationship [28]. Water resources are a major production factor for sustainable socioeconomic development, and thus there are many studies of the relationship between total water usage and economic growth [29,30]. China is a country with very short water resources. The continuous growth of GDP and total water consumption makes the impact of water resource constraints on economic growth critical [30]. China's extensive economic development model has resulted in several problems related to water resources, including low efficiency of use, waste, and pollution. Therefore, water resources represent an input variable for sustainable economic development.

Construction land is a resource factor that is closely linked, and highly contributes to economic growth [31]. Throughout the rapid urbanization and industrialization, China's large-scale urban construction and economic activities of all kinds have been accompanied by the constant expansion of construction land. Therefore, area of construction land is treated as a resource factor in the process of economic growth and is incorporated into the input indicator system.

At the same time, China's entire course of economic development has been accompanied by massive energy consumption, with the country surpassing the United States in 2010 as the world's largest consumer of energy. However, while energy consumption can be a powerful driver of economic growth [32], excessive energy consumption can cause a plethora of environmental problems that restrict its speed and quality [33]. Energy consumption is therefore selected as an important input indicator.

### 2.2. Socioeconomic Factors

Fixed asset investment and the number of employed people are used to represent capital and labor factors affecting economic growth. Investment is an important engine of economic development and fixed asset investment is closely connected to the development of the national economy [34]. Assuming the amount of resources and level of technological sophistication are fixed, the speed of economic growth is largely determined by the total size and growth of investment [35]. Many studies show that fixed-asset investment is strongly correlated with real national output, while the positive or negative results of investment can affect the economy's operational efficiency [34,35]. The number of employed people, on the other hand, is one way to reflect the labor market, and is both influenced by economic growth and can simultaneously be a significant indicator of growth trends [36].

In addition, as countries are paying increasing attention to raising the quality of development, the role of science and technology as a contributor and driver of economic growth is becoming increasingly prominent, providing an endogenous impetus to economic growth. It has become a key factor spurring the transformation of growth models [37]. Therefore, the intensity of research and development expenditure, which is abbreviated as R&D, is adopted as an important indicator measuring the extent of a region's input into science and technology.

### 2.3. Environmental Factors

China is one of the world's major industrial countries, and its economic growth has been accompanied by pervasive environmental pollution over the last four decades. Industrial pollution accounts for 70% of the country's total pollution load [38]. Although the Environmental Kuznets Curve, abbreviated as EKC theory, posits a bell-shaped curve between the level of pollutants discharged and economic development, its increase could restrict the speed of economic development to a significant degree [39]. Existing studies mostly deal with environmental factors as input variables [40] or undesired output variables [17]. Referring to the former approach, typical industrial pollution (solid industrial waste, liquid industrial waste, gaseous industrial waste) is therefore used as the environmental input. The less pollutant emissions in economic activities, the lower the cost of controlling pollutants, which means less input and higher economic output efficiency.

**Table 2.** Input-output efficiency indicator system of economic growth.

| Variable Type | Category | Indicator |
|---|---|---|
| Input variable | Natural resource factor | Total water use<br>Built up area<br>Total energy consumption |
| | Socioeconomic factor | Fixed asset investment<br>Total number of employed people<br>Intensity of R&D expenditure |
| | Environmental factor | Total solid industrial waste<br>Total liquid industrial waste<br>Total gaseous industrial waste |
| Output variable | Economic output | GDP |

## 3. Methods and Data

### 3.1. Methods

Under the premise of hypothesizing the different efficiency frontier shapes as well as the distribution of random errors and efficiency values, previous studies have introduced efficiency research methods based on "frontier analysis". Data envelopment analysis (DEA) is just a method used for frontier analysis [41]. DEA is a statistical method for measuring the relative validity or effectiveness of decision-making units in multiple equivalent categories that have multiple inputs and outputs. It is an ideal model for quantifying input-output efficiency [41–43], which has been widely used since 1978 to evaluate the relative efficiency of multi-input and multi-output production units [42]. The advantage of this method is that it determines the weights of various input factors endogenously using the optimization method, which avoids the need for a specifically expressed relationship between input and output and eliminates a large number of subjective factors. Within the DEA model, every economic entity or production process with a definite yield through the input of production factors is called a decision-making unit (DMU). The essence of the model is using the weight of all the DMUs' input-output indicators as variables for evaluation and calculation. First, the efficient production frontiers are determined, and then it is decided whether each DMU is effective, depending on its distance from the frontier.

With the traditional DEA model, the effectiveness of a DMU is determined by the dichotomous "effective" or "ineffective", their assessed value being unity and less than unity, respectively, making collation and ranking difficult if not impossible. Andersen and Petersen [44] proposed an improved model in 1993, which was super-efficiency DEA, based on the traditional DEA model. This model serves the defect that the traditional DEA model cannot further compare the efficient decision-making units, and can identify the ranks of effective DMUs through frontier conversion [44]. This was employed in our study, as illustrated in Figure 2.

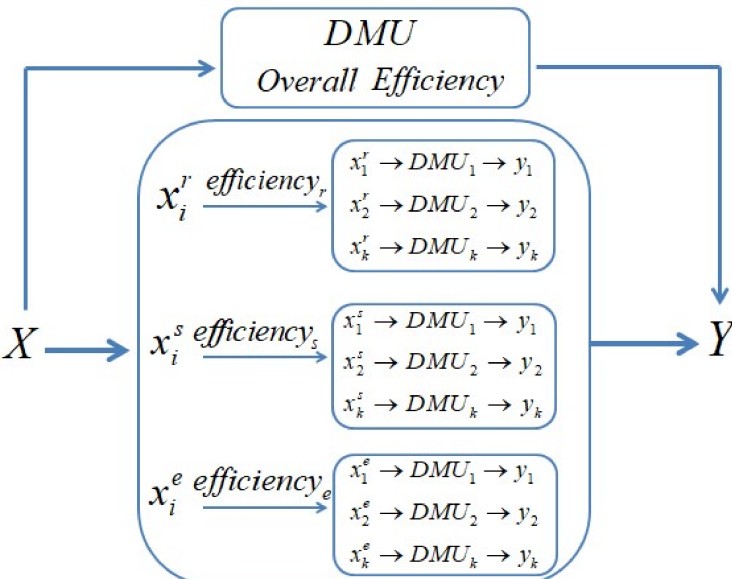

**Figure 2.** Analytical framework based on the super-efficiency data envelopment analysis (DEA) model. Note: $X$ represents the set of input factors and $Y$ represents the output factor, which is GDP in this study. $x_i^r, x_i^s, x_i^e$ represent the input elements of natural resources, social economy, and environmental pollution of the *i*-th DMU, respectively. $y_i$ represents the GDP of the *i*-th DMU. *Efficiency$_r$*, *efficiency$_s$*, *efficiency$_e$* represent resource efficiency, socioeconomic efficiency, and environmental efficiency, respectively.

Concerning existing studies [19], this article employed the radial super-efficient DEA model. Supposing that the input-output efficiency of $n$ regions (DMUs) will be evaluated, every DMU has $m$ input variables and $s$ output variables, $x_{ik}$ indicates input variable $i$ in region $k$, $y_{jk}$ indicates output variable $j$ in region $k$, then the formula for calculating the input-output efficiency of DMU $k$ is as follows [26]. The efficiency values were achieved via DEA-SOLVER Pro 5.0 software. We noticed that the traditional DEA model is only applicable to cross-sectional data, which would be used to compare the status of each decision-making unit at one time point. In applying the DEA model, we measured the input-output efficiency of 30 provinces year by year, which was still based on the application of cross-sectional data. Then, we used the multi-year average of the efficiency results to characterize the spatial and temporal patterns of efficiency. To reflect the dynamic change trend characteristics of each decision-making unit's efficiency in the time series, we adopted relative value comparison thinking, and ranked the DEA efficiency values of all decision-making units each year. Furthermore, we analyzed the temporal change trend of input-output efficiency in various regions by observing the ranking changes of provinces in different years, and we strove to avoid incomparable absolute efficiency values over many years.

$$X_k = (x_{1k}, x_{2k}, \ldots, x_{mk}), \ Y_k = (y_{1k}, y_{2k}, \ldots, y_{sk}) \tag{1}$$

$$\min\theta$$

$$(CCR)\,s.t. \begin{cases} \sum_{\substack{j=1 \\ j\neq k}}^{n} X_j\,\lambda_j \leq \theta X_k \\ \sum_{\substack{j=1 \\ j\neq k}}^{n} Y_j\,\lambda_j \geq Y_k \\ \lambda_j \geq 0, j = 1,2,\dots,n \end{cases} \tag{2}$$

### 3.2. Study Area

The research area covered 30 provinces of China's mainland (excluding Hong Kong and Macao). The Tibet Autonomous Region was not considered because of insufficient indicators and data. At the same time, to depict the input-output efficiency of regional economic growth in China more clearly from a macro-level perspective, the 31 provinces were divided into eastern, northeastern, central, and western macro-regions, following the regional division strategy proposed during the 11th Five-Year Plan period (Figure 3).

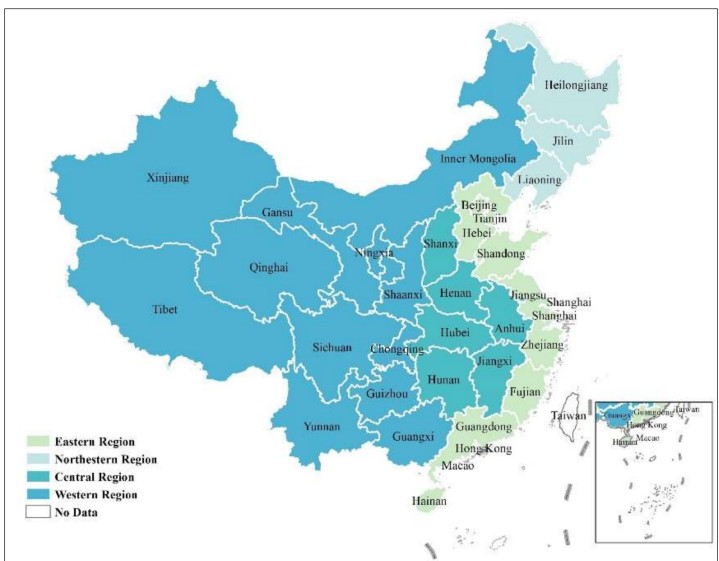

**Figure 3.** The four major regions of China.

### 3.3. Data

The socioeconomic, environmental, and resource data for each province cited were primarily sourced from the China Statistical Yearbook (2001–2016) and the statistical yearbooks of each province, with some supplementary data provided by professional statistical yearbooks. The area of construction land was sourced from the China Urban Statistical Yearbook (2001–2016), total water usage from the China Water Resources Bulletin (2000–2015), total energy consumption from the China Energy Statistical Yearbook (2001–2016). Data for the amounts of solid, liquid, and gaseous waste produced by provinces, regions, and cities were mostly taken from the China Environmental Statistics Yearbook (2001–2016) with supplementation from the Winds information database and National Development and Reform Commission statistical data. Taking the data of 2000 and 2015 as representatives, a statistical description indicator data set is displayed (Table 3).

**Table 3.** Variables' descriptive statistics in this study in 2000 and 2015.

| Year | Statistics | Total Water use (100 Million Cubic Meters) | Built-Up Area (km²) | Total Energy Consumption (1000 Tons of Standard Coal) | Total Solid Industrial Waste (10 Thousand Tons) | Total Gaseous Industrial Waste (Billion Standard Cubic Meters) | Total Liquid Industrial Waste (10 Thousand Tons) | Intensity of R&D Expenditure (%) | Total Number of Employed People (Ten Thousand People) | Fixed Asset Investment (100 Million Yuan) | GDP (100 Million Yuan) |
|---|---|---|---|---|---|---|---|---|---|---|---|
| | mean | 181.68 | 540.70 | 4868.77 | 2720.43 | 4604.33 | 64,727.04 | 0.87 | 2226.77 | 2909.72 | 3283.19 |
| | std | 122.89 | 345.53 | 2715.84 | 2057.46 | 2954.28 | 47,973.14 | 0.93 | 1530.85 | 9837.40 | 2548.04 |
| 2000 | CV | 0.68 | 0.64 | 0.56 | 0.76 | 0.64 | 0.74 | 1.08 | 0.69 | 3.38 | 0.78 |
| | Minimum | 23.02 | 61.00 | 480.00 | 95.00 | 434.00 | 4661.00 | 0.15 | 275.50 | 154.83 | 264.00 |
| | Maximum | 460.40 | 1443.00 | 10,766.00 | 7695.00 | 12,179.00 | 201,923.00 | 4.92 | 5572.00 | 54,820.00 | 10,741.25 |
| | mean | 202.16 | 1319.03 | 14,621.77 | 10,889.31 | 23,075.70 | 66,483.33 | 1.63 | 2787.06 | 18,505.05 | 24,083.81 |
| | std | 148.56 | 849.14 | 8566.04 | 9505.50 | 17,126.77 | 53,039.13 | 1.16 | 1831.68 | 11,588.42 | 18,038.21 |
| 2015 | CV | 0.73 | 0.64 | 0.59 | 0.87 | 0.74 | 0.80 | 0.71 | 0.66 | 0.63 | 0.75 |
| | Minimum | 24.10 | 103.00 | 1937.77 | 422.00 | 2338.70 | 6879.00 | 0.46 | 316.19 | 3210.60 | 2417.10 |
| | Maximum | 591.30 | 3597.00 | 36,759.20 | 35,372.00 | 78,570.00 | 206,427.00 | 6.03 | 6636.08 | 48,312.40 | 72,812.60 |

## 4. Results

### 4.1. Spatiotemporal Evolution of Input-Output Efficiency

#### 4.1.1. Overall Efficiency

The results show the spatial differentiation of input-output efficiency in 2015 (Figure 4), with 12 provinces having overall efficiency values exceeding unit, thereby reaching DEA effectiveness. These included 10 provinces from the eastern region, Inner Mongolia from the western region, and Henan from the central region. Beijing's overall efficiency was much higher than other provinces at 3.167. At 1.657, 1.559, 1.405, and 1.292, respectively, Guangdong, Shanghai, Shandong, and Tianjin all had overall efficiency values significantly higher than unity, while those of Hebei, Hainan, and Henan were slightly lower. None of the overall input-output efficiency values of the 18 other provinces reached DEA effectiveness. Liaoning, Qinghai, Heilongjiang, and Xinjiang had efficiency values between 0.9 and 1.0, and were the closest to achieving DEA effectiveness. Hunan, Yunnan, Shanxi, Jiangxi, Guizhou, and Sichuan in central and western China had efficiency values between 0.8 and 0.9, while Guangxi, Chongqing, Jilin, Shaanxi, Hubei, and Anhui had values between 0.7 and 0.8, placing them in the mid to lower ranks countrywide. Gansu and Ningxia ranked the lowest in the country, with efficiency values below 0.7. In regional terms, the overall efficiency levels were ranked as eastern region > northeastern region > central region > western region.

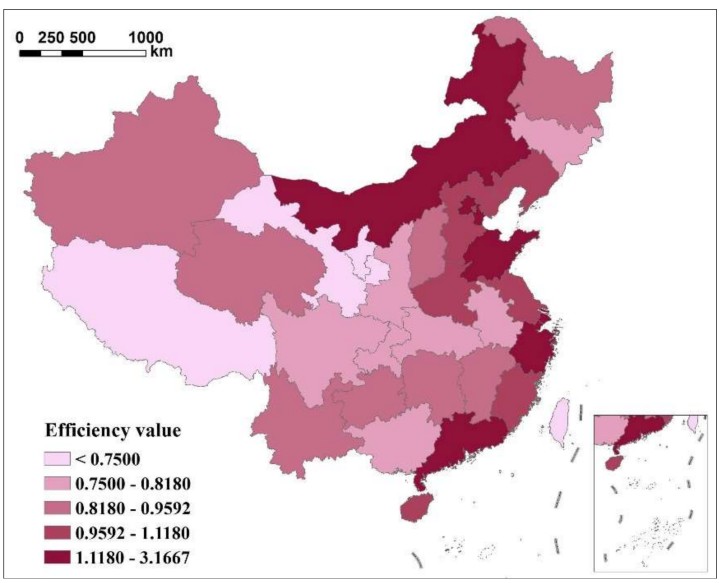

**Figure 4.** Spatial differentiation of overall input-output efficiency.

In terms of evolutionary trends (Figure 5), most eastern provinces had the highest overall efficiency levels and were relatively stable. For example, Beijing, Guangdong, and Shanghai remained in the top five provinces with the country's overall efficiency for many years, while Shandong, Tianjin, and Zhejiang steadily maintained their top 10. Hebei and Jiangsu jumped up in rank quite significantly, with Hebei rising from 21st place in 2000 to 12th before reaching 7th in 2013, and Jiangsu rising from 12th place in 2000 to 5th in 2015. In contrast, the overall efficiency rankings of Fujian and Hainan fell substantially over successive years. In the northeastern region, Liaoning has climbed up in recent years, while Jilin and Heilongjiang's rankings continue to fall. The central provinces of Henan, Hunan, and Hubei rose somewhat in the national overall efficiency rankings between 2000 and 2015, with Hubei jumping noticeably in 2015. The rankings of Shanxi and Anhui fell markedly, indicating the overall efficiency levels of these two provinces to be subpar from a national perspective. The overall efficiency ranking of Jiangxi rose significantly between 2003 and 2012 but then fell from

2012 onward, rising from 26th to 25th place. The western provinces of Inner Mongolia, Sichuan, Guizhou, Shaanxi, and Qinghai rose noticeably in the overall efficiency rankings. Yunnan rose slightly, Guangxi, Chongqing, and Xinjiang fall on the whole, and Gansu and Ningxia constantly remained at the bottom of the rankings.

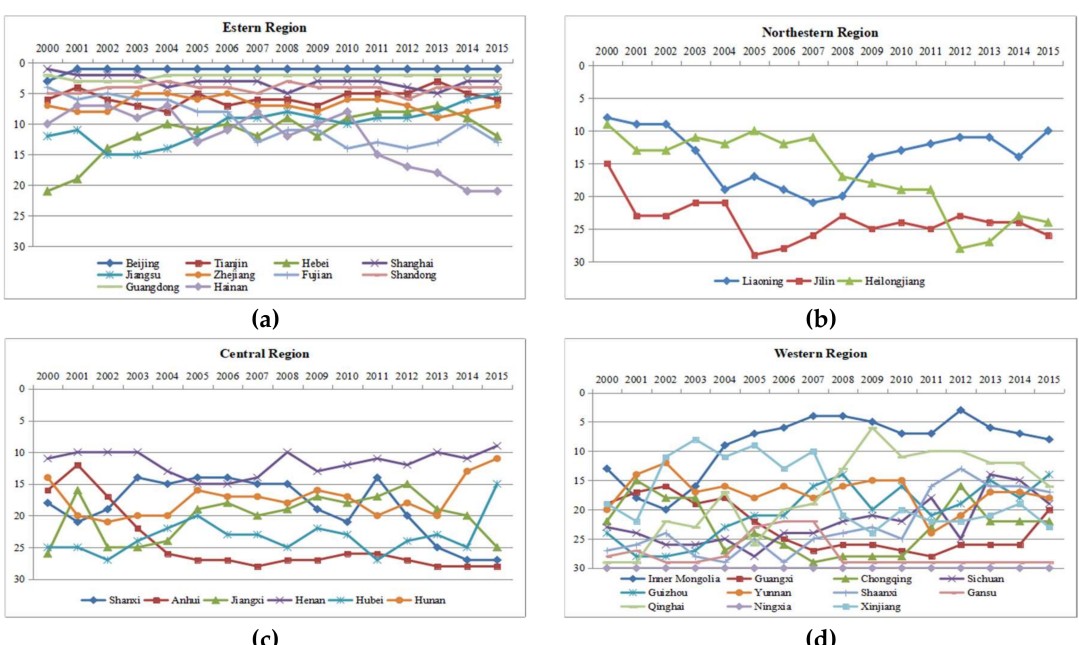

**Figure 5.** Changes of overall input-output efficiency rankings. of Eastern (**a**), Northeastern (**b**), Central (**c**), and Westren (**d**) regions in China from 2000 to 2015.

### 4.1.2. Resource Efficiency

Resource efficiency refers to the relative quantity of socioeconomic benefits produced per unit of natural resource factor input. The spatial differentiation of resource efficiency in 2015 showed the provinces with high resource efficiency levels to be Beijing, Tianjin, Fujian, Shanghai, and Zhejiang, with efficiency values of 1.403, 1.173, 1.069, 1.057, and 1.034, respectively—all of which achieved DEA effectiveness. The provinces with relatively high resource efficiency levels (0.9-1.0) were Guangdong, Shandong, and Jiangsu; while those with medium levels (0.8-0.9) were Hebei, Hainan, Qinghai, and Jiangxi; and provinces with relatively low levels (0.7-0.8) were Yunnan, Henan, Xinjiang, Shanxi, Hunan, Shaanxi, and others. Those in the low-level category were mainly in the central and western regions. The majority of the remaining provinces all had efficiency values below 0.7, and were therefore quite far from achieving DEA effectiveness. Ranked the lowest for efficiency levels were the three northeastern provinces as well as Inner Mongolia, Gansu, and Ningxia. Overall, regional resource efficiency was ranked as eastern region > central region > western region > northeastern region (Figure 6).

Between 2000 and 2015 (Figure 7), eastern China had resource efficiency levels that were ranked the highest overall, the rankings of its provinces being nearly stable. For example, Beijing's resource efficiency was ranked first for many years. The rankings of Shanghai, Jiangsu, Zhejiang, and Guangdong experienced different degrees of fluctuation, but have risen in recent years. Tianjin jumped from seventh place in 2000 to second in 2015. The rankings of Hebei, Fujian, Shandong, and Hainan fell somewhat, with Fujian and Hainan experiencing the most significant drop. The three northeastern provinces took up the rear in resource efficiency and experience different degrees of decline. Liaoning fell from 20th place in 2000 to 27th in 2015, while the rankings of Heilongjiang and Jilin fell somewhat amid fluctuations and settled at 28th and 24th, respectively, in recent years. In the central region, Hunan and Henan significantly increased in their rankings, while those of Shanxi, Anhui, Jiangxi, and Henan fell

by different amounts. Different western provinces experienced relatively large fluctuations in their resource efficiency rankings, with Guizhou and Qinghai having the most obvious increases, while the rankings of Chongqing and Xinjiang fell at first and then rose. Those of Inner Mongolia and Shaanxi first rose and then fell. For example, the resource-rich province of Inner Mongolia increased its ranking from 28th to 15th between 2000 and 2012, and then fell to 23rd in 2015. Besides, the resource efficiency rankings of provinces, including Guangxi, Sichuan, and Yunnan, fluctuated and fell to an extent, while Gansu and Ningxia were consistently ranked at the bottom.

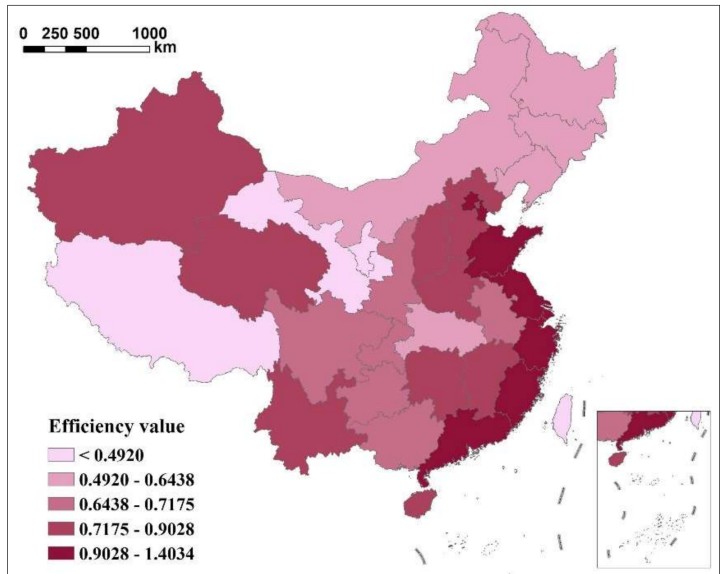

**Figure 6.** Spatial differentiation of resource efficiency.

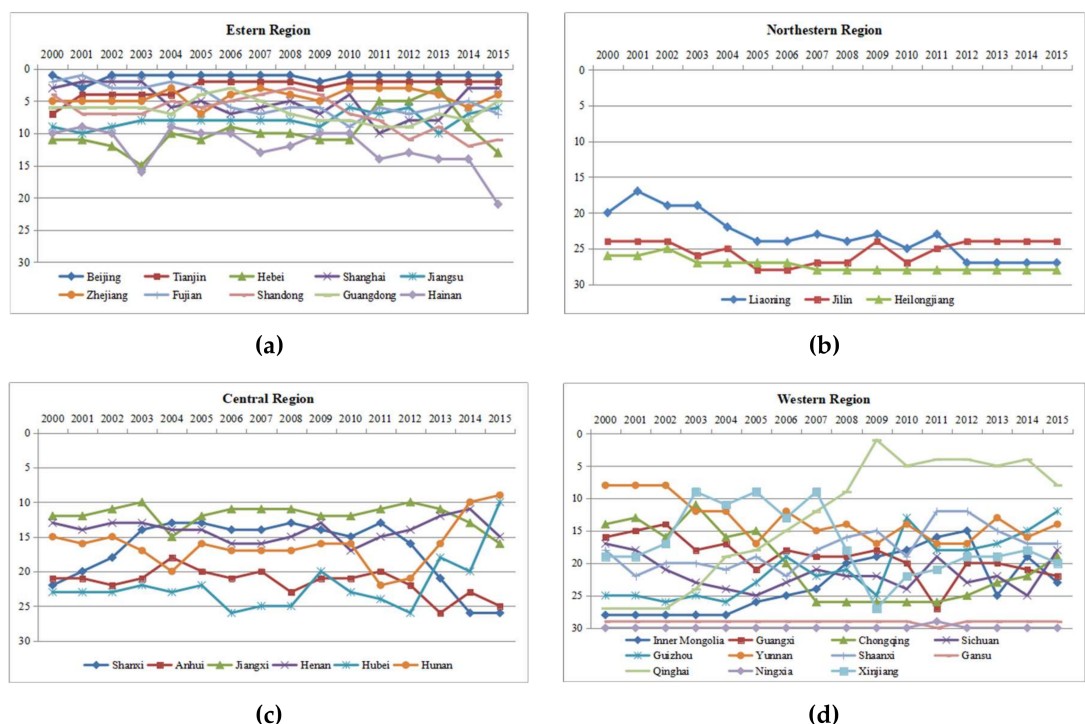

**Figure 7.** Changes of resource efficiency rankings. of Eastern (**a**), Northeastern (**b**), Central (**c**), and Westren (**d**) regions in China from 2000 to 2015.

### 4.1.3. Socio-Economic Efficiency

As shown in Figure 8, in 2015, only Shanghai and Guangdong reached DEA effectiveness in socioeconomic efficiency, with values of 1.515 and 1.414, respectively. Areas with relatively high socioeconomic efficiency levels (0.9-1.0) comprised Henan, Jiangsu, Shandong, Beijing, Tianjin, Inner Mongolia, and Hebei; those with medium levels (0.8-0.9) comprised Zhejiang, Fujian, and Liaoning; and those with relatively low levels (0.7-0.8) comprised Heilongjiang, Guangxi, and Hunan. The remaining 15 provinces were all areas with low socioeconomic efficiency, with efficiency values lower than 0.7. At the regional level, the spatial layout of socioeconomic efficiency was similar to that of overall efficiency: eastern region > northeastern region > central region > western region. Furthermore, for the vast majority of China's provinces, there is tremendous potential for the input-output efficiency of socioeconomic factors to increase.

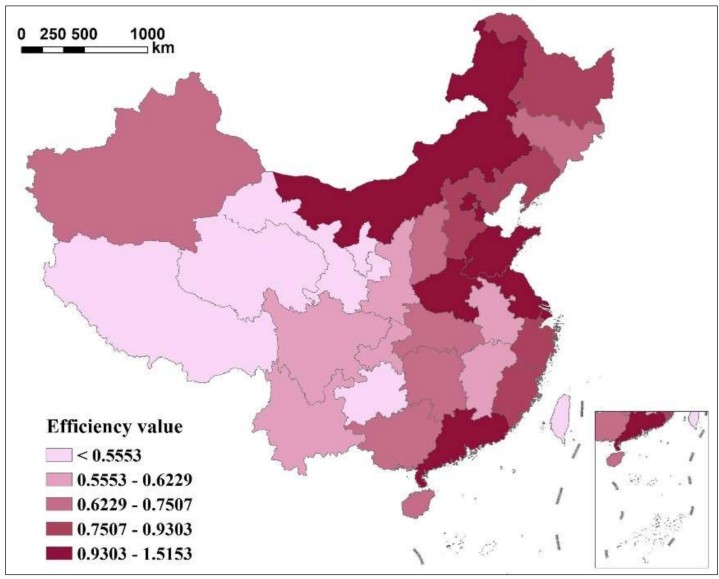

**Figure 8.** Spatial differentiation of socioeconomic efficiency.

Figure 9 shows that between 2000 and 2015, the socioeconomic efficiency rankings of Beijing, Tianjin, Hebei, and Jiangsu in the eastern region rose significantly, while those of Zhejiang, Fujian, Shandong, and Hainan fell; and Shanghai and Guangdong consistently ranked relatively high. In the northeastern region, Heilongjiang's ranking fell, while Liaoning and Jilin fell slightly but were essentially stable. In the central region, Henan took the lead in socioeconomic efficiency rankings with a slight increase, while the ranks of the other five provinces fell to different degrees, with Anhui experiencing the most remarkable drop. Most western provinces rose in rank, but not by a huge margin. Inner Mongolia was the western province with the highest socioeconomic efficiency, ranked third in the nation, but its rank fell in the last few years. Guangxi and Chongqing first fell and then rose in rank relatively quickly overall. The rankings of Sichuan, Guizhou, Shaanxi, Qinghai, and Xinjiang rose slightly through a fluctuation, and those of Yunnan, Gansu, and Ningxia fell.

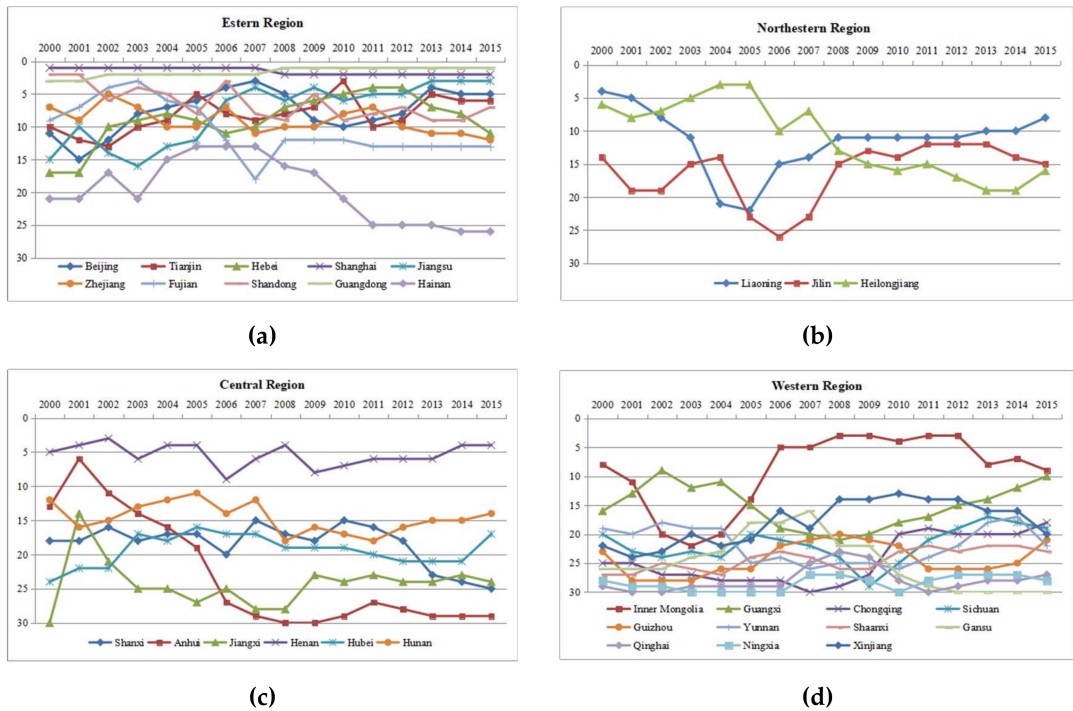

**Figure 9.** Changes in socioeconomic efficiency rankings. of Eastern (**a**), Northeastern (**b**), Central (**c**), and Westren (**d**) regions in China from 2000 to 2015.

### 4.1.4. Environmental Efficiency

The environmental efficiencies were markedly lower than the input-output efficiency levels of the two preceding subsystems, indicating there is some friction between economic development and the environment (Figure 10). Only Beijing's environmental efficiency reached DEA effectiveness, with a value of 3.151; none of the 29 other provinces met the threshold. Guangdong was in the relatively high range of environmental efficiency (0.8-0.9); Hainan, Shanghai, Zhejiang, Tianjin, and Fujian in the middle range (0.5-0.8); and the Jiangsu, Shandong, and Hebei provinces in the eastern region had low environmental efficiency, especially Hebei, which ranked third-last in the nation. All the provinces in the central region, northeastern region, and western region had efficiency values below 0.5. Furthermore, most provinces in the central region and northeastern region had environmental efficiency values of between 0.3 and 0.4, while most western provinces had values below 0.3. The regional environmental efficiency rankings were eastern region > central region > northeastern region > western region.

Between 2000 and 2015 (Figure 11), the environmental efficiency rankings of the eastern provinces changed quite significantly. Beijing's level was the highest in the nation throughout this period, and Guangdong's ranking was also stable and remained in the top three. Meanwhile, the rankings of Tianjin, Shanghai, and Zhejiang had an upward trend. Hainan's ranking fell slightly; Jiangsu, Fujian, and Shandong fell significantly; and Hebei's ranking was both low and with a downward trend. In the northeastern region, Jilin and Heilongjiang's rankings rose noticeably, while Liaoning's also rose but to a lesser extent. In the central region, Hunan, Shanxi, and Henan fluctuated only slightly and essentially maintained their rankings. Hunan had a comparatively high level, and steadily held on to its position in the top ten. Hubei's ranking rose markedly from 23rd in 2000 to 14th in 2015, while Anhui and Jiangxi's rankings dropped by more than ten. In the western region, the rankings of Inner Mongolia, Guangxi, Chongqing, Sichuan, Guizhou, Shaanxi, and Gansu all increased, while those of Yunnan, Qinghai, and Xinjiang fell significantly. The drop was unusually large for Xinjiang, which fell from 9th in 2000 to 25th in 2015. Meanwhile, Ningxia was consistently ranked last in the country for environmental efficiency.

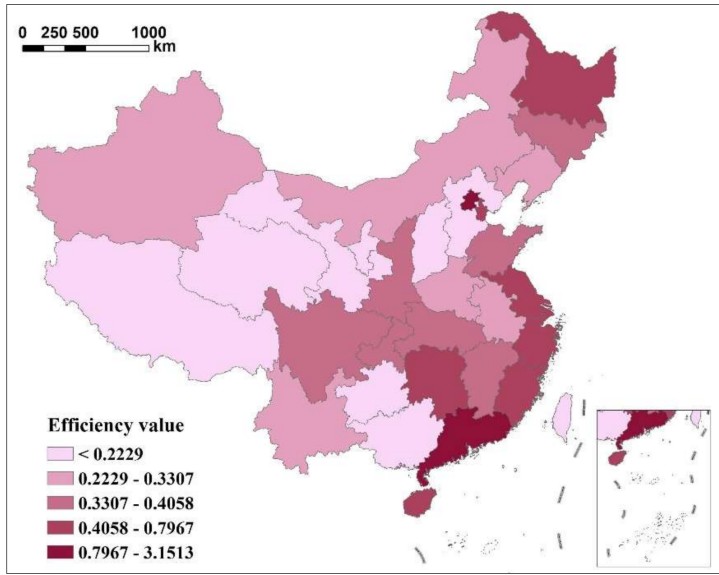

**Figure 10.** Spatial differentiation of average environmental efficiency.

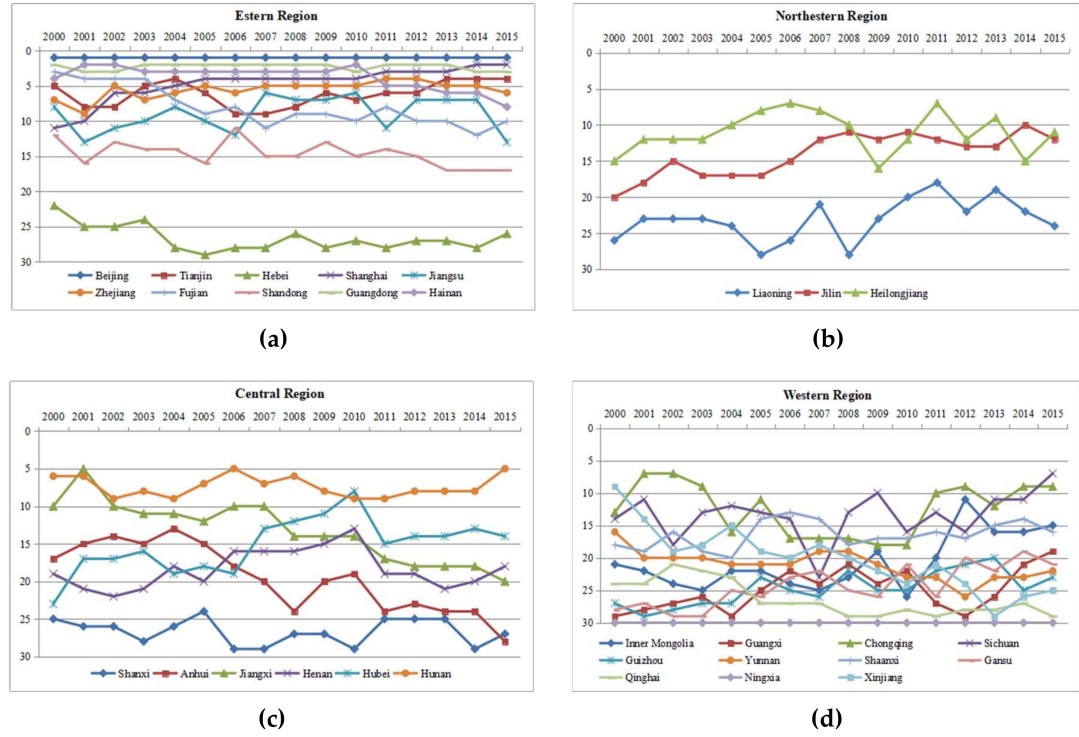

**Figure 11.** Changes in environmental efficiency rankings. of Eastern (**a**), Northeastern (**b**), Central (**c**), and Westren (**d**) regions in China from 2000 to 2015.

*4.2. Constraint Analysis and Categorization of Input-Output Efficiency*

Table 2 provides the results from categorizing input-output efficiency based on the bisection method, with ranks 1-10, 11-20, and 21-30 representing relatively high level, medium, and low levels, respectively. These results indicate that provinces in eastern China all rank in the medium or above for overall efficiency and socioeconomic efficiency, and in the high range for resource efficiency, but are spread across the high, medium, and low ranges for environmental efficiency. As a result, Eastern China should pay more attention to raising environmental efficiency and improve its economic quality. The provinces in northeastern China are ranked in the medium and low ranges for overall efficiency,

and in the medium range for socioeconomic efficiency. The environmental efficiency levels vary significantly between provinces, while resource efficiency levels are all low. This shows that vigorously enhancing the input-output efficiency of resource utilization is the key for northeastern provinces to increase their economic efficiency. The provinces in central China are mainly ranked in the medium and low ranges for overall efficiency, with their resource efficiency rankings generally matching those for overall efficiency. The socioeconomic and environmental efficiency levels are also relatively low, which indicates the potential for improvement in input-output efficiency on the resource, socioeconomic, and environmental levels. The low overall input-output efficiency levels are concentrated in the provinces of western China, as are the low resource, socioeconomic, and environmental efficiency levels. Socioeconomic efficiency has a notable role in restricting overall efficiency levels. Therefore, highly efficient economic development in western China faces challenges on multiple levels: including further increasing the efficiency of resource use, raising the level of economic development, enhancing the effectiveness of economic output, and reducing environmental pollution.

The "wooden barrel theory" stresses the importance of optimizing all of a system's functions [45]. According to this theory, if the efficiency of any subsystem is insufficient, then overall input-output efficiency will be impacted and constrained by the low-efficiency subsystem, which will produce a "bottleneck" effect on regional economic efficiency. Therefore, it is possible to identify low-efficiency subsystems within the input-output efficiency systems by combining the overall input-output efficiency of each province with subsystem efficiency, including resource efficiency, socioeconomic efficiency, and environmental efficiency of different provinces (Table 4). This information can be used to determine the main constraining factors on the overall efficiency of different provinces, based on which we can identify the different types of input-output efficiency-restricted provinces.

**Table 4.** Comparison of regional subsystem efficiency levels.

| Region | DMU | Overall Efficiency | | Resource Efficiency | | Socioeconomic Efficiency | | Environmental Efficiency | |
|---|---|---|---|---|---|---|---|---|---|
| | | Relative Level | Rank | Relative Level | Rank | Relative Level | Rank | Relative Level | Rank |
| Eastern Region | Beijing | H | 1 | H | 1 | H | 6 | H | 1 |
| | Tianjin | H | 5 | H | 2 | H | 7 | H | 6 |
| | Hebei | H | 10 | H | 9 | H | 9 | L | 28 |
| | Shanghai | H | 3 | H | 4 | H | 1 | H | 4 |
| | Jiangsu | H | 9 | H | 8 | H | 4 | H | 9 |
| | Zhejiang | H | 6 | H | 5 | H | 10 | H | 5 |
| | Fujian | H | 8 | H | 3 | M | 12 | H | 7 |
| | Shandong | H | 4 | H | 7 | H | 5 | M | 15 |
| | Guangdong | H | 2 | H | 6 | H | 2 | H | 2 |
| | Hainan | M | 11 | H | 10 | M | 20 | H | 3 |
| Northeastern Region | Liaoning | M | 13 | L | 25 | M | 11 | L | 23 |
| | Jilin | L | 25 | L | 27 | M | 16 | M | 14 |
| | Heilongjiang | M | 15 | L | 28 | M | 13 | H | 10 |
| Central Region | Shanxi | M | 19 | M | 16 | M | 18 | L | 29 |
| | Anhui | L | 28 | L | 22 | L | 23 | M | 19 |
| | Jiangxi | M | 20 | M | 12 | L | 24 | M | 11 |
| | Henan | M | 12 | M | 14 | H | 3 | M | 18 |
| | Hubei | L | 27 | L | 24 | M | 19 | H | 8 |
| | Hunan | M | 17 | M | 17 | M | 15 | H | 8 |
| Western Region | Inner Mongolia | H | 7 | L | 26 | H | 8 | L | 22 |
| | Guangxi | L | 23 | M | 19 | M | 14 | L | 26 |
| | Chongqing | L | 24 | M | 20 | L | 26 | M | 12 |
| | Sichuan | L | 22 | L | 23 | L | 22 | M | 13 |
| | Guizhou | L | 21 | L | 21 | L | 27 | L | 25 |
| | Yunnan | M | 18 | M | 13 | L | 21 | L | 21 |
| | Shaanxi | L | 26 | M | 18 | L | 25 | M | 17 |
| | Gansu | L | 29 | L | 29 | L | 28 | L | 24 |
| | Qinghai | M | 14 | M | 11 | L | 30 | L | 27 |
| | Ningxia | L | 30 | L | 30 | L | 29 | L | 30 |
| | Xinjiang | M | 16 | M | 15 | M | 17 | M | 20 |

(1) Resource efficiency-constrained

There are 15 resource efficiency-constrained provinces, most of which are located in the northeastern, western, and central regions (Figure 12a). The three northeastern provinces have relatively poor resource efficiency, and thus, the northeastern region is a typical example of a region constrained by resource efficiency. Western China, Inner Mongolia, Shaanxi, Sichuan, Gansu, and Ningxia are also areas with significant resource efficiency constraints. The central region, Anhui, Henan, Hubei, and Hunan are all subject to the constraining effect of resource efficiency. In the eastern region, although resource efficiency levels are relatively high, Shanghai, Jiangsu, and Guangdong are all constrained by resource efficiency to some extent.

(2) Socioeconomic efficiency-constrained

There are 14 socioeconomic efficiency-constrained provinces, most of which are in the western and central regions (Figure 12b). The western provinces are subject to the most obvious constraints from socioeconomic efficiency. These include Chongqing, Sichuan, Guizhou, and Yunnan in the southwest, and Gansu, Qinghai, and Ningxia in the northwest. The central provinces of Anhui and Jiangxi are also subject to relatively clear constraints from socioeconomic efficiency. In the eastern region, Beijing, Tianjin, Zhejiang, Fujian, and Hainan are also subject to some degree of constraint from socioeconomic efficiency.

(3) Environmental efficiency-constrained

There are 19 environmental efficiency-constrained provinces, most of which are in the western, central, and eastern regions (Figure 12c). The provinces in the western region are significantly constrained by environmental efficiency, with Inner Mongolia, Shaanxi, Guangxi, Guizhou, Yunnan, Gansu, Qinghai, Ningxia, and Xinjiang all ranked as the most affected. The central provinces of Shanxi, Henan, Anhui, and Hubei, as well as the northeastern province of Liaoning, are also constrained by environmental efficiency. These provinces are all important industrial bases for energy and raw materials in China, and industrial pollution severely restricts their environmental efficiency. The eastern provinces that are significantly constrained by environmental efficiency include Hebei and Shandong, while the rankings of Tianjin, Shanghai, and Jiangsu in environmental efficiency lag behind their rankings in other subsystems.

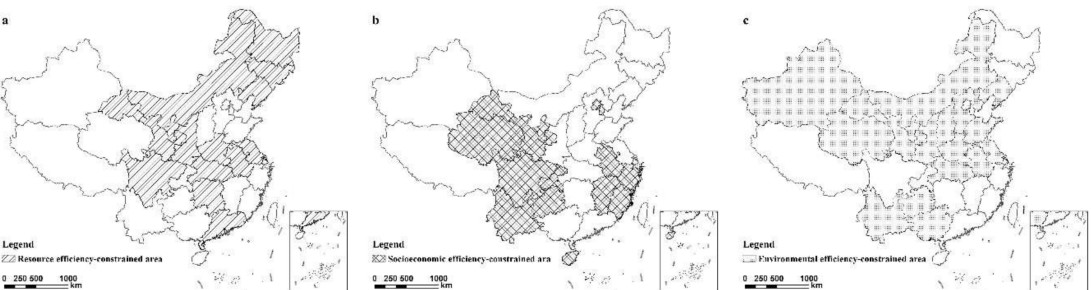

**Figure 12.** Spatial distribution of different constrained areas., including resource efficiency-constrained provinces (**a**), socioeconomic efficiency-constrained provinces (**b**), and environmental efficiency-constrained provinces (**c**).

## 5. Discussion

By exploring the efficiency of input-output systems from a holistic perspective, most previous studies considered only the system's overall inputs and outputs while neglecting the relationship between different subsystems. By carrying out a comparative analysis of the input-output levels of various subsystems, this study presents a more complete examination of the characteristics of the country's spatiotemporal evolution of regional input-output efficiency and identification of the factors restricting increased economic efficiency. Moreover, it provides policy implications for improving regional economic efficiency as follows.

(1) Resource efficiency-constrained provinces: the three northeastern China provinces of Inner Mongolia, Sichuan, and Gansu; Ningxia in the western region; and Anhui, Henan, Hubei, and Hunan in central China are the least resource efficiency constrained. As a traditional industrial base, the northeastern region has historically had an overdependence on resources and heavy industry, and a "path dependency" concerning reliance on resource and energy consumption through large-scale industrial development. This has caused low resource efficiency to become a significant constraining factor on their input-output efficiency in the process of economic growth. In the future, it will be necessary for the northeastern region to transform its modes of economic development, upgrade industrial structures, and expand traditional industrial chains to shift from the output of raw materials to refined processed products. Meanwhile, the western region has gradually entered a stage of industrialization and accelerated development since the launch of China's major national strategy to develop the west of the country. Its resource consumption has constantly grown in intensity. Western China is also an economically underdeveloped region, where the initial processing of resources has always dominated the industry. For example, many industries in Gansu and Ningxia are technologically backward and have high energy and resource demands. These factors have combined to make many western provinces low resource efficient. The central region does not have a clear advantage in input-output efficiency for resource factors despite its abundance of natural resources, with the supply of such production factors like land and energy decreasing in Anhui, Henan, Hubei, and Hunan. The resource efficiency bottleneck is becoming increasingly apparent, particularly as resource extraction has become more costly and challenging, and resource use has remained inefficient. The eastern region provinces have relatively high resource efficiency levels overall. However, Shanghai, Jiangsu, and Guangdong are constrained by resource efficiency to a significant degree, as they are developed provinces and industrial and urban development have brought them close to the limit of their carrying capacity in terms of such resource and environmental factors as water, land, and energy.

(2) Socioeconomic efficiency-constrained provinces: The western region is notably constrained by socioeconomic efficiency, including the provinces of Chongqing, Sichuan, Guizhou, and Yunnan in the southwest; and Gansu, Qinghai, and Ningxia in the northwest. These provinces have relatively lagging economic conditions, inefficient fixed asset investments, and limited scientific and technological resources, which has made their socioeconomic input-output efficiency low overall. The Anhui and Jiangxi provinces in the central region are also constrained by socioeconomic efficiency, as they have relatively slow economic development rates and are dominated by energy-intensive industries. They urgently need further adjustment and optimization of investment structures centered on upgrading their industrial structures and enhancing their capacity of scientific and technological innovation. In the eastern region, Beijing, Tianjin, Zhejiang, and Hainan are constrained by socioeconomic efficiency. Beijing, Tianjin, and Zhejiang have already reached comparatively high levels of industrialization and urbanization, so their ability to boost economic growth by inputting socioeconomic factors is quite limited. In the future, they will need innovation to drive a transformation in the efficiency of the socioeconomic factor input. Hainan, as an economically underdeveloped province in the eastern region, should focus on the quality and effectiveness of its socioeconomic factor input to raise the output efficiency of all input factors. For the provinces constrained by socioeconomic efficiency, those that are economically underdeveloped need to further promote the input of socioeconomic factors that spur economic growth, while those that are already economically developed should explore new sources of momentum for factor input and form new drivers for economic growth.

(3) Environmental efficiency-constrained provinces: Most provinces in the western region are significantly constrained by environmental efficiency. This is especially the case with Inner Mongolia, Guangxi, Guizhou, Yunnan, Gansu, Qinghai, Ningxia, and Xinjiang. They are dominated by highly polluting traditional processing and manufacturing industries that use outdated production techniques and are ill-equipped for environmental protection. As a result, high-intensity solid, liquid, and gaseous pollutants are emitted and their environmental efficiency is low. Liaoning in the northeast and Shanxi, Henan, Anhui, and Hubei in the central and western regions are all subject to environmental efficiency

constraints. These provinces are important industrial bases for energy and raw materials in China, with large energy-intensive and highly polluting industries involved in such activities as coal extraction and processing, steel, and nonferrous metals. Therefore, industrial pollution severely constrains their environmental efficiency. In the future, these provinces need to increase their determination to conserve energy and cut emissions, actively promote clean production, and focus on forming economic and high-efficiency growth models featuring low input, energy consumption, and emissions. In the eastern and western regions, Hebei and Shandong are subject to relatively significant environmental efficiency constraints. For Hebei, this is a result of the province's singular industrial structure, in which such heavy industries as steel, petrochemicals, and building materials are dominant. Meanwhile, Shandong has a large population but comparatively small environmental capacity, and has experienced the rapid growth of its highly polluting industries in such activities as papermaking, oil extraction, and chemical manufacture. Besides, Tianjin, Shanghai, and Jiangsu have relatively low environmental efficiency rankings; this indicates that, although these three provinces are well developed, they still need to pay serious attention to environmental protection in the course of their economic growth.

Although this paper provides new enlightenment for the study of input-output efficiency and a series of targeted policy recommendations, there are still some limitations. First, this paper prioritizes the feasibility of the realization of the research objectives. Based on the existing data, only the traditional super-efficiency DEA model was used to measure the input-output efficiency level of a region for the consideration of research objectives, macroscopic scientific issues, feasibility, and applicability instead of bogging down to the technical levels. DEA, a linear programming model, is not the only method to measure efficiency. There are many methods for efficiency evaluation. For example, the stochastic frontier analysis, which is abbreviated as SFA, another commonly used efficiency evaluation method in academic circles, is a typical representative of parametric methods in frontier analysis [46]. Its main advantage is that it can consider the impact of random factors on output. The two methods have their advantages and disadvantages, and the measurement results of the same problem may be different. Therefore, it is worth exploring the comparative analysis of SFA and DEA. If necessary, the combination of the measurement results of the two methods will be a topic worthy of further study in the future. Second, this paper's current research process tended to regard the efficiency level of 30 decision-making units as independent, without considering the spatial dependence (i.e., spatial interaction) between the research units, and the traditional geostatistical thinking was preferred. However, many studies have shown that [18,47] there is a particular spatial relationship between regions in input-output efficiency. In the future, spatial measurement analysis methods can be further introduced to study the spatial effect of input-output efficiency among regions in China based on detailed research units. Furthermore, the spatial econometric analysis method can be introduced to further study the spatial effect of input-output efficiency among regions in China and whether there is a spatial relationship between input-output activities among regions based on detailed research units. In addition, in the design of input-output efficiency indicators, such as environmental factors, this paper used different types of pollutant emissions to represent the environmental impact, without the impact of ecosystem service functions on economic growth and input-output activities. In the future, the indicator system can be further improved by taking into account the factors of ecosystem services, which provide an impetus for social and economic development. It could also provide more and richer reference information for measuring regional input-output efficiency and formulating sustainable development policies.

## 6. Conclusions and Policy Implications

Previous studies have been unable to provide a full picture of actual input-output characteristics, which makes it challenging to identify the location of bottlenecks in regional input-output efficiency. Based on a study of 30 Chinese provinces from 2000 to 2015, this paper carried out a comprehensive evaluation of overall input-output efficiency using a super-DEA model, and decomposed the input indicator system and calculated resource, socioeconomic, and environmental efficiency to identify

each province's bottleneck factors. We mainly found that most eastern provinces had the highest overall efficiency, with Gansu and Ningxia constantly remaining at the bottom. Liaoning, Henan, Hunan, and Hubei rose in the rankings over the years, while Heilongjiang, Jilin, Shanxi, and Anhui fell markedly. Most eastern provinces had the highest resource efficiency, while Gansu and Ningxia consistently ranked the lowest. Guizhou, Qinghai, Hebei, Fujian, Shandong, Hainan, Liaoning, and Jilin all experienced different degrees of decline, while Guizhou, Qinghai, Hunan, and Hebei increased in the rankings. Most eastern provinces stayed at a high level of socioeconomic efficiency, while Anhui, Jiang, Qinghai, Hainan, and Ningxia ranked the lowest. Most western provinces rose in rank, while the central provinces except Henan fell. The environmental efficiencies were markedly lower than the levels of another two subsystems. Most western and northeastern provinces increased in rank, while most eastern and central provinces fell. Finally, we think that China's provinces can be divided into three categories of resource, socioeconomic, and environmental efficiency-constrained. There were 15 resource efficiency-constrained provinces, most of which were located in the northeastern, western, and central regions; 14 socioeconomic efficiency-constrained provinces, most of which were from the western and central regions; and 19 environmental efficiency-constrained provinces, most of which were from the western, central and eastern regions.

The above findings provide a series of meaningful policy implications and indicate typical values to similar countries out of China.

For the resource efficiency-constrained provinces, the natural resource investment is high and the efficiency is low, due to the heavy industry or resource primary processing industry dominating structure. Industrial transformation is the inevitable choice for these provinces to achieve sustainable development. As the initiator and promoter of industrial transformation, the government has formulated industrial transformation policies, and put various production factors (such as financial, material, and human resources) into appropriate regions, to foster the development of industrial transformation. Besides, the rational utilization of resources and the rational allocation of the distribution system should have importance attached, to provide strong financial support for promoting the industrial transformation as well as the development of new industries in resource-efficiency-restricted areas. In addition, these areas should introduce or establish high-tech enterprises, and increase the total output value and added value of high-tech industries, to integrate existing innovation resources and optimize unreasonable industrial structures.

For socioeconomic efficiency-constrained provinces, some central and western provinces have low socioeconomic input-output performance efficiency because of economically backwardness. These provinces must coordinate the relationship between total economic growth and efficiency improvement by shifting the growth pattern from the expansion of low-cost investment to an innovation-driven and efficiency-driven pattern. They should also pay attention to the improvement of economic growth efficiency and the cost of resources and environment. Moreover, they should establish a green and low-carbon development concept, focus on energy saving and emission reduction, and strengthen the adjustment of industrial structure by "retreating the secondary industry into the tertiary industry". On the other hand, some eastern provinces, where the socio-economic input factors have driven the economy to a greater extent, should pay more attention to accelerating science and technology advancement, improving independent innovation capabilities, and increasing the share of technology, innovation, and knowledge. In terms of the policy, the practice of intervening in industrial development should be abolished. It is necessary to create more advanced production factors, improve the quality of demand, encourage new businesses and preserve market competition. At the same time, these provinces should thoroughly learn from the experience of developed countries, strictly formulate and implement energy conservation and emission reduction standards, to encourage the exploration and development of new processes and technologies, and cultivate the innovation potential of growth.

For environmental efficiency-constrained provinces, various unfavorable outputs should be well dealt with, such as reducing the pollutant discharge of carbon dioxide, wastewater, waste residue, etc., and the abilities of government and enterprises for coping with environmental pollution should

be enhanced. The upgrading of the industry is from structural adjustment, technological progress, and improved management. These provinces should strive to achieve "removing pollution from the source" in production, guide the rational allocation of production resources, and improve the efficiency of green development. Besides, it is not appropriate to implement environmental regulation by "imposing a single solution". For industries with competitive advantages in developed provinces, increasing environmental protection investment and improving environmental infrastructure is essential. At the same time, these regions should actively explore new ways of pollution control in the new situation. It is essential to establish a diversified investment mechanism consisting of governments, enterprises, and society, a market-oriented operation mechanism for some pollution control facilities, and a unified, coordinated, and effective environmental supervision system.

**Supplementary Materials:** The following are available online at http://www.mdpi.com/2071-1050/12/11/4624/s1.

**Author Contributions:** Conceptualization, Z.S. and L.K.; methodology, L.K.; software, L.K.; validation, Z.S.; formal analysis, L.K.; investigation, L.K.; resources, Z.S. and L.K.; data curation, Z.S. and L.K.; writing—original draft preparation, L.K.; writing—review and editing, Z.S.; visualization, L.K.; supervision, Z.S.; project administration, Z.S.; funding acquisition, Z.S. and L.K. All authors have read and agreed to the published version of the manuscript.

**Funding:** This research was funded by the National Natural Science Foundation of China (Grant No. 41430636, 41801114).

**Conflicts of Interest:** The authors declare no conflict of interest.

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
