# Peer review of "Input-Output Efficiency of Economic Growth: A Multielement System Perspective"

_sustainability, doi:10.3390/su12114624_

Round 1

Reviewer 1 Report

Input-Output Efficiency of Economic Growth: A Multielement System Perspective

Introduction

The introduction is well written. However, I do not know the background of the current efficiency level of the economic growth of China. Again, the introduction should capture the current economic growth of China. Moreover, the authors presented a review of literature on methodology used by previous studies without outlining the key findings from the studies. It is important to report the findings from the study beside the methodology employed. Lines 33 to 39 need to be referenced. The contributions of the study should be more explicit in the introduction.

Research Framework

Section 2 should be reframed as “Conceptual framework”

Lines 93 to 106 need to be referenced. Please provide the source of the conceptual framework in Figure 1. If its your own design then reference it as Authors’ design (2020). Lines 138 to 140 require references.

Methods and Data

Lines 163 to 165 require references.  Provide references for lines 168-172. Please place the title of Figure 2 right below it.

3.2 Case study  should be reframed as “The study area”.

Please provide references for lines 192 to 198.

Please provide summary statistics and some graphs to describe the data set under the data section.

The conclusion section is poorly written. The authors should take their time to highlight the key policy implications of their findings.

Author Response

Response to reviewers

Many thanks for the insightful comments and suggestions concerning our manuscript (ID: Sustainability-797181). We have considered your comments and have incorporated almost all your suggestions in the revised version of our paper. We also have a detailed response to each comment in the below. In the paper, the changes made are clearly highlighted using the "Track Changes" function. We feel that by revising our paper according to your constructive suggestions, the paper has been greatly improved.

The following are the answers and revisions.We have made in response to the reviewers' questions and suggestions on an item by item basis.

Reply to Referee 1

Specific Comments 1

The introduction is well written. However, I do not know the background of the current efficiency level of the economic growth of China. Again, the introduction should capture the current economic growth of China.

Moreover, the authors presented a review of literature on methodology used by previous studies without outlining the key findings from the studies. It is important to report the findings from the study beside the methodology employed.

Lines 33 to 39 need to be referenced.

The contributions of the study should be more explicit in the introduction.

Response to comment 1:

Thanks a lot to these helpful suggestions. We have revised and improved the article one by one according to the reviewer’s comments.

1) We have added sufficient content about China’s status. The supplementary content includes both the introduction of the study area in the first draft and our new explanation. Please check the revised version in Line 127-151.

2) We have added the key findings of the the main literature. And we will pay attention to avoid such problems in the future.

3) We have added the relevant citations, including the references [1-3]

4) We have added a lot of content in the Introduction including the discussion and comments on existing literature, and improvement of perspective of our study, in order to stress the contributions of this study. Please check the revised version in Line 55-61, Line 100-120, and Line 158-163.

Specific Comments 2

Section 2 should be reframed as “Conceptual framework”

Lines 93 to 106 need to be referenced. Please provide the source of the conceptual framework in Figure 1. If its your own design then reference it as Authors’ design (2020).

Lines 138 to 140 require references.

Response to comment 2:

1)We have renamed Section 2 as “Conceptual framework”. Please check the revised version in Line 170.

2)After combing and analyzing the existing research progress, we have designed the conceptual framework presented in the article according to the research goals, in order to enrich relevant research and the research perspectives.So the conceptual framework was designed by ourselves, which is also our first attempt to publish. We are sorry that we can't find a suitable citation for the time being. Thank you a lot for your understangding and supporting us.

3)All supporting references for the selection of indicators have been added. Please check the revised version in Line 195-242.

Specific Comments 3Methods and Data

Lines 163 to 165 require references. Provide references for lines 168-172.

Please place the title of Figure 2 right below it.

3.2 Case study should be reframed as “The study area”.

Please provide references for lines 192 to 198.

Please provide summary statistics and some graphs to describe the data set under the data section.

Response to comment 3:

1) We have added the citation about the method, including the references[42-45].

2) We have placed the title of Figure 2 right below it. Please check the revised version in Line 269.

3) We have reframed Case study as “The study area”.

4) We have provided references about status of China’s economic and sustainable development. Please check the revised version in Line 131-142.

5) According to the suggestion, taking the space limitations and data representativeness into account, we have displayed a statistical description indicator data set of 2000 and 2015 as representatives. Please check Table 3 in the revised version.

Specific Comments 4Conclusion

The conclusion section is poorly written. The authors should take their time to highlight the key policy implications of their findings.

Response to comment 4:

Thanks a lot for your valuable comments. According to your suggestion, we have tried our best to enrich the conclusion, and added Section 6.2(Policy implications) and Section 6.3(Limitations). Please check the revised version.

Reviewer 2 Report

Dear authors,

Regarding this article proposal I have the following questions.

Please answer each question separately.

1)

On the methodology side I have three questions

i)

My first question is why the DEA was chosen, in relation to the Panel Data Analysis, because you have enough data to address the economic and social phenomenon iChina by region, and years?

ii)

It is also useful to compare the results with another popular alternative such as Stochastic Frontier Analysis (SFA) approach.

Please find heredetails and discussions in Chapters 8, 9 and 10 for DEA and Chapters 11 through 16 for SFA in Sickles, R., & Zelenyuk, V. (2019). Measurement of Productivity and Efficiency: Theory and Practice. Cambridge: Cambridge University Press. doi:10.1017/9781139565981 https://assets.cambridge.org/97811070/36161/frontmatter/9781107036161_frontmatter.pdf

Resource is free.

Why did you choose DEA with “frontier analysis”  and not DEA with Stochastic Frontier Analysis  methodology?

iii)

I noticed that maps were generated, the third question is why space econometrics was not used, because the literature offers many methods and resources for spatial analysis?

2) Note on the data side

As it is an open journal in this case, please provide the analyzed data source.

A statistical yearbook has been provided, but readers will probably want to view this data processed by you.

Eventually upload on a platform the data indicating the DOI source.

3)  Software

The article did not specify what software was used to process the data using DEA.

Please indicate the software application, possibly its version.

Respectfully

Calin-Adrian COMES

Author Response

Response to reviewers

Many thanks for the insightful comments and suggestions concerning our manuscript (ID: Sustainability-797181). We have considered your comments and have incorporated almost all your suggestions in the revised version of our paper. We also have a detailed response to each comment in the below. In the paper, the changes made are clearly highlighted using the "Track Changes" function. We feel that by revising our paper according to your constructive suggestions, the paper has been greatly improved.

The following are the answers and revisions.We have made in response to the reviewers' questions and suggestions on an item by item basis.

Reply to Referee 2

Specific Comments 1

My first question is why the DEA was chosen, in relation to the Panel Data Analysis, because you have enough data to address the economic and social phenomenon in China by region, and years? Response to comment 1:

Thanks to your suggestion focusing on the method in this article. The opinion leads us to further review and think about the process of applying this method to this research. We chose the DEA method, mainly because this method is very suitable for our research goal. The calculation process is to measure the input-output efficiency of 30 provinces year by year, and the calculation process by year also meets the DEA method requirements for sample size. Based on our current understanding of the DEA, we know that DEA is a statistical method for measuring the relative validity or effectiveness of decision-making units in multiple equivalent categories that have multiple inputs and outputs, and is an ideal model for quantifying input-output efficiency. The research aims to characterize the spatial-temporal pattern of efficiency based on multiple input-output indicators of economic development in China. So the DEA model is applicable.

Regarding the panel data mentioned in the suggestion, although it is not the direct reason for the DEA method used in this article, we must admit that the observations of the indicators do constitute panel data with multiple cross-sections (30 administrative units) ,and a certain period of time (2000-2015).Then because we noticed that the traditional DEA model is only applicable to cross-sectional data, which means it would be used to compare and analyze the status of each decision-making unit at one time point. In the process of applying DEA model, we measure the input-output efficiency of 30 provinces year by year, still based on the application of cross-section data. In order to characterize the spatial and temporal pattern of efficiency, we use the multi-year average of the efficiency results. In order to reflect the dynamic change trend characteristics of the efficiency of each decision-making unit in the time series, we adopt relative value comparison thinking. It means we rank the DEA efficiency values of all decision-making units in each year, and analyze the temporal change trend of input-output efficiency in various regions by ovserving the ranking changes of provinces in different years, and strive to avoid incomparable absolute efficiency values over many years.

We are very sorry that there was no detailed and clear description about DEA's usage ideas in the first draft. It has now been supplemented in section 3.1.Please check the revised version in Line 280-291.

Specific Comments 2

It is also useful to compare the results with another popular alternative such as Stochastic Frontier Analysis (SFA) approach.

Please find heredetails and discussions in Chapters 8, 9 and 10 for DEA and Chapters 11 through 16 for SFA in Sickles, R., & Zelenyuk, V. (2019). Measurement of Productivity and Efficiency: Theory and Practice. Cambridge: Cambridge University Press. doi:10.1017/9781139565981 https://assets.cambridge.org/97811070/36161/frontmatter/9781107036161_frontmatter.pdf. Resource is free.

Why did you choose DEA with “frontier analysis” and not DEA with Stochastic Frontier Analysis methodology?

Response to comment 2:

First of all, we are very grateful to the very inspiring suggestions from the perspective of professional methods and techniques, and also to the riviewer for providing relevant learning materials and detailed sources. As far as we know, SFA is a typical representative of the parameter method in frontier analysis, and the specific form of the production front needs to be determined. Its main advantage is the impact of random factors on output could be considered. While the DEA is a non-parametric method with measuring efficiency through linear programming.It does not need to know the specific form of the production frontier. Only the input and output data is enough. DEA is convenient to deal with the decision-making unit with multi-input and multi-output. In comparison, SFA's assumptions are more complex and require higher input-output data. If the input-output data does not meet the basic assumptions, it will easily lead to calculation failure. Based on the existing data and materials, the DEA method has been used in this paper to meet the research goals. At the same time, we are so sorry that our understanding of SFA is still not deep enough, so the suggestions are of great help to us to further expand the horizon and depth of efficiency issues. Inspired by expert opinions, we have known that the two methods have their own advantages and disadvantages, and the measurement results for the same problem may be different. It is worth exploring the comparative analysis of the two methods of SFA and DEA in the future. Maybe the measurement results by the two methods could be integrated.

As for why choose DEA with “frontier analysis” and not DEA with Stochastic Frontier Analysis methodology, it can be seen that there are not too many technical considerations for the selection of research methods in our study, combining with the content of the article and response to suggestion 1. The reason why we choose the traditional DEA model is more related to the specific research objectives of this article,along with the feasibility and applicability of the method. To be practical, there is still much room for improvement in the application of the method. Fortunately, the results obtained can well reflect the problems at different levels in the sustainable development of China and different regions. On the one hand, the results are consistent with China's development reality and people's experience and cognition.On the other hand, it reveals the characteristics of the spatial and temporal evolution of China's regional economic development efficiency from a system perspective, and tries to find out the restrictive factors to economic efficiency, making people's understanding of China's sustainable development problems more comprehensive and richer.Previous studies have paid more attention to the comprehensive investigation of input-output efficiency, and failed to dismantle the efficiency evaluation of complex systems like our article. This is the core concern of this study.

In summary, this article may pay more attention to the setting of research ideas and the enrichment and expansion of research perspectives. And we are very sorry that due to the limited time, no more in-depth research has been done on methods except for traditional DEA models. We agree a lot with the reviewer's attention to the research method, as well as the helpful comments on it. The discussion of methods by experts has a good guiding significance for further improvement and deepening of research in the future, especially the comparative study around DEA and SFA methods, which is a topic worthy of in-depth study. Relevant instructions have been added in the last part of conclusion section. And we have also added relevant literature citation as reference[48]. Please check the revised version in Line 693-704.

Specific Comments 3

I noticed that maps were generated, the third question is why space econometrics was not used, because the literature offers many methods and resources for spatial analysis?

Response to comment 3:

Thanks very much for the deep and detailed review of this article. In our study, we tend to regard the efficiency of the 30 decision-making units as independent, and does not consider the spatial dependence between the units, which means the spatial interaction between the units is not taken into consideration based on the traditional geostatistical thinking. The maps generated in this article are intended to reflect the spatial pattern and differences of China's various input-output efficiencies, but cannot reflect the spatial interaction between regions. In the future, the spatial econometric analysis methods can be further introduced to conduct further in-depth research on the spatial effect of input-output efficiency between regions, so as to identify whether there is a spatial connection among regional input-output activities. Relevant instructions have been added in the last part of conclusion section. Please check the revised version in Line 705-714.

Specific Comments 4

As it is an open journal in this case, please provide the analyzed data source.

A statistical yearbook has been provided, but readers will probably want to view this data processed by you.

Eventually upload on a platform the data indicating the DOI source.

Response to comment 4:

According to the suggestion, we have uploaded the original data and the result data.

Specific Comments 5

The article did not specify what software was used to process the data using DEA.

Please indicate the software application, possibly its version.

Response to comment 5:

Thanks to the suggestion. It is our negligence for not introducing the software and version used for data processing. The efficiency value were achieved via DEA-SOLVER Pro 5.0 software. Please check the revised version in Line 281.

Reviewer 3 Report

Comments to the authors

  1. p.1, ln14. "always" is a very strong expression and I think this is little exaggerated and recommend the authors to remove this word.
  2. p.2, ln41, "only way" is an exaggerated expression. I don't think this is the only way.
  3. p.2, ln43-44. The use of the word "important" is repetitive.
  4. p.2, ln56. Although it is a well-known abbreviation in the relevant field, the paper should write the full term for SBM when using the abbreviation for the first time in the paper.
  5. p.2, ln78. More discussions and explanations should be provided to support the reasons for focusing the study to these three subsystems.
  6. p.2, ln80. “In this study…” Sentences after this sentence should better to be put in a new paragraph. The reason for applying the analyses to China should be explained.
  7. Introduction section lacks enough evidence to support the significance of the study. More discussions and broader literature should be included to show why this study is important. The section should also discuss what are the expected outcomes from the study.
  8. p.3, ln107-109. A literature should be included to support this sentence. Or the authors should provide in a quantitative way like providing the number of studies focusing on every aspect.Including a table organizing the literature with their publication years, names of authors, methods and the content of the research, and so on would be ideal.
  9. p.3, ln119. “China’s extensive model...” As already suggested, more extensive discussions should be done to explain why the paper focused the case study for China before this sentence.
  10. p.4, ln134-41, “Fixed asset…of growth trends.” para. References to support this assumption must be cited.
  11. p.4, ln143-148, para. Again, references to support this assumption must be cited.
  12. p.5, ln178, super-efficiency DEA model. This is the main model of the study but the current explanation of the method is very shallow. First, some history of the method starting from Andersen and Petersen (1993) needs to be introduced. Second, there are several models within the super-efficiency DEA such as the radial and non-radial models. More detailed selection of the model used in the paper must be explained.
  13. p.6, Section 3.2. This type of explanation about choosing China as the case study should have been introduced in the earlier section of the paper.
  14. p.5-7, Methods section. For the readers to replicate the study, the paper should provide the statistical software package or tools used for the analyses.
  15. p.16 Conclusions section. This section should provide what implications can be driven from the findings of the study rather than simply listing the findings of the study. Furthermore, the paper did not provide the limitation of the study, which is a recommended by the journal. This could be added in the discussions section too.

Author Response

Response to reviewers

Many thanks for the insightful comments and suggestions concerning our manuscript (ID: Sustainability-797181). We have considered your comments and have incorporated almost all your suggestions in the revised version of our paper. We also have a detailed response to each comment in the below. In the paper, the changes made are clearly highlighted using the "Track Changes" function. We feel that by revising our paper according to your constructive suggestions, the paper has been greatly improved.

The following are the answers and revisions.We have made in response to the reviewers' questions and suggestions on an item by item basis.

Reply to Referee 3

Specific Comments 1

p.1, ln14. "always" is a very strong expression and I think this is little exaggerated and recommend the authors to remove this word.

Response to comment 1:

Thanks a lot for this suggestion. We are very sorry that the expression of “always” is really not precise and rigorous enough. We have removed this word, and changed it to “tend to”. Please check the revised version in Line 14.

Specific Comments 2

p.2, ln41, "only way" is an exaggerated expression. I don't think this is the only way.

Response to comment 2:

We are sorry again that the "only way" is an exaggerated expression. Thanks for your suggestion. We changed it to “an important way”. Please check the revised version in Line 41.

Specific Comments 3

p.2, ln43-44. The use of the word "important" is repetitive.

Response to comment 3:

Thanks for your suggestion. We have changed the first “important” to “a research topic of great concern”. Please check the revised version in Line 44.

Specific Comments 4

p.2, ln56. Although it is a well-known abbreviation in the relevant field, the paper should write the full term for SBM when using the abbreviation for the first time in the paper.

Response to comment 4:

Thanks to the reviewer for the suggestions on the rigorous language expression of the article. According to this suggestion, we added the full term for SBM, which is called “Slack Based Model”. Please check the revised version in Line 64.

Specific Comments 5

p.2, ln78. More discussions and explanations should be provided to support the reasons for focusing the study to these three subsystems.

Response to comment 5:

Thanks to this suggestion. Please check the revised version in Line 104-120.We have added discussions and introduction shown in red as follows:

However, the extensive studies of regional input-output efficiency, although helpful in improving environmental protection and sustainable development, mostly regard regional input-output efficiency as a completely undivided system. And the mainly providing provide only an overall evaluation based on a series of input-output indicators that is unable to analyze the properties of the structure and components within the system. In fact, the economic activities involve the consumption of various factors, which means the input-output factor system involved is very complicated. In order to avoid the imbalance of the evaluation system due to excessive emphasis on certain factors, the input-output efficiency evaluation often selects a series of elements as input indicators in order to pursue the comprehensiveness of the evaluation process, and then conducts a comprehensive evaluation. This makes it easy to obscure the effect characteristics of input factors in some aspects, so that the evaluation results would stay at the system level and cannot penetrate to the element level. In reality, there can be obvious discrepancies between the efficiency levels of subsystems obtained by dividing regional input-output efficiency according to the different input factors involved, with different subsystems having their own efficiency characteristics. Previous comprehensive studies usually fail to monitor the efficiency levels of each subsystem separately from the element type level, and making it difficult challenging to identify the bottlenecks in input-output efficiency. Therefore, it is necessary to characterize the input-output efficiency by different subsystems to reflect the differences and effects of different types of elements, and then analyze the problems existing in the entire economic system . It is conducive to minimizing resource input and environmental losses in the process of economic output and maximizing economic benefits.

Specific Comments 6

p.2, ln80. “In this study…” Sentences after this sentence should better to be put in a new paragraph. The reason for applying the analyses to China should be explained.

Response to comment 6:

Thanks to the suggestions. Firstly, we have put “In this study” in a new paragraph. Besides, we have added sufficient content about reason for applying the analyses to China. The supplementary content includes both the introduction of the study area in the first draft and our new explanation. Please check the revised version in Line 127-151.

Specific Comments 7

Introduction section lacks enough evidence to support the significance of the study. More discussions and broader literature should be included to show why this study is important. The section should also discuss what are the expected outcomes from the study.

Response to comment 7:

Thanks to the suggestions. Your opinion is very helpful to highlight the research significance of this article.

We have added more discussions and broader literature to support the significance of the study. Please check the revised version in Line 55-61, Line 79-99.

And the expected outcomes from the study have been added. Please check the revised version in Line 158-163.

Specific Comments 8

p.3, ln107-109. A literature should be included to support this sentence. Or the authors should provide in a quantitative way like providing the number of studies focusing on every aspect.Including a table organizing the literature with their publication years, names of authors, methods and the content of the research, and so on would be ideal.

Response to comment 8:

Thanks to this suggestion. According to the reviewer’s suggestion, we have added a summary of eco-efficiency research object, methods and indicators in the form of a table marked with references. Please check Table1 in Line 191 in the revised version.

Specific Comments 9

p.3, ln119. “China’s extensive model...” As already suggested, more extensive discussions should be done to explain why the paper focused the case study for China before this sentence.

Response to comment 9:

Thanks to this suggestion. We have added some introduction about China with citation. Please check the revised version in Line 199-201.

Specific Comments 10

p.4, ln134-41, “Fixed asset…of growth trends.” para. References to support this assumption must be cited.

Response to comment 10:

Thanks to this suggestion. Relevant citations have been added, referring to the literature[34] and [35].

Specific Comments 11

p.4, ln143-148, para. Again, references to support this assumption must be cited.

Response to comment 11:

Thanks to this suggestion. Relevant citations have been added, referring to the literature[36] and [37].

Specific Comments 12

p.5, ln178, super-efficiency DEA model. This is the main model of the study but the current explanation of the method is very shallow. First, some history of the method starting from Andersen and Petersen (1993) needs to be introduced. Second, there are several models within the super-efficiency DEA such as the radial and non-radial models. More detailed selection of the model used in the paper must be explained.

Response to comment 12:

Thanks to this suggestion. According to the reviewer’s opinion, we have added some history of the method. Plese check the revised version in Line 251-253, and Line 263-266.

Secondly, we explaine that this article employed the radial super-efficient DEA model with reference to existing studies[19]. Plese check the revised version in Line 276-277.

Specific Comments 13

p.6, Section 3.2. This type of explanation about choosing China as the case study should have been introduced in the earlier section of the paper.

Response to comment 13:

According to the reviewer’s suggestion, we have moved the introduction about China to the earlier section. Plese check the revised version in Line 127-145.

Specific Comments 14

p.5-7, Methods section. For the readers to replicate the study, the paper should provide the statistical software package or tools used for the analyses.

Response to comment 14:

Thanks to this suggestio. It is really our negligence for not introducing the software and version used for data processing. The efficiency values were achieved via DEA-SOLVER Pro 5.0 software. Please check the revised version in Line 280.

Specific Comments 15

p.16 Conclusions section. This section should provide what implications can be driven from the findings of the study rather than simply listing the findings of the study. Furthermore, the paper did not provide the limitation of the study, which is a recommended by the journal. This could be added in the discussions section too.

Response to comment 15:

Thanks a lot for your valuable comments. According to the reviewer’s suggestion, we have tried our best to enrich the conclusion, with changing the Conclusion Section to Conclusion and policy implications Section. Please check the revised version in Line 632-680. Besides, we have added the explanation pointing out the limitation of the study which can be checked in Line 682-713.

Round 2

Reviewer 1 Report

The authors have addressed all my concerns. However, I suggest that the authors combine the conclusions and policy implication as one main section. The conclusion still needs be revised. Please, do not number the conclusion.

Author Response

Many thanks for the further comments concerning our manuscript (ID: Sustainability-797181). First of all, thank you for your recognition of the previous round of revision. This time we have considered your comments in the revised version of our paper. We also have a detailed response to each comment in the below. In the paper, the changes made are still clearly highlighted using the "Track Changes" function. The following are the answers and revisions. We have made in response to the reviewers' questions and suggestions on an item by item basis.

Comments

The authors have addressed all my concerns. However, I suggest that the authors combine the conclusions and policy implication as one main section. The conclusion still needs be revised. Please, do not number the conclusion.

Response:

Thanks a lot for this suggestion. In the revised part, we have combined the conclusions and policy implication as one main section, which is Section 6 “Conclusions and policy implications”. And we have removed the number of conclusions. Besides, we have moved the limitation analysis to Section 5 “Discussion”. Please check the revised version (Line 630-703).

Reviewer 3 Report

Thank you for having me the opportunity to review your paper. Overall the authors did very well to revise the paper according to my comments but I feel the paper should provide a good explanation why the paper focused on the three efficiencies: resource efficiency, socio- economic efficiency, and environmental efficiency.

  1. It is still not clear why the study focuses on resource efficiency, socio- economic efficiency, and environmental efficiency.
  2. There are still many typos like ln121, "stduy."
  3. A space is missing before the brackets for the reference. For example, in ln128, grown[23] should be grown [23].
  4. ln665. SFA is perhaps the stochastic frontier analysis but this is not explained anywhere in the paper. Again the authors avoid using abbreviations without providing the full word somewhere in the text.
  5. A minor english proof editing is necessary before the final submission.

Author Response

Response to reviewers

Many thanks for the further comments concerning our manuscript (ID: Sustainability-797181). First of all, thank you for your recognition of the previous round of revision. This time we have considered your comments in the revised version of our paper. We also have a detailed response to each comment in the below. In the paper, the changes made are still clearly highlighted using the "Track Changes" function. The following are the answers and revisions. We have made in response to the reviewers' questions and suggestions on an item by item basis.

Specific Comments 1

It is still not clear why the study focuses on resource efficiency, socio-economic efficiency, and environmental efficiency.

Response: Thanks for your valuable comments. First, according to the UNDP Report, the realization of the UN 2030 Sustainable Development Goals mainly depends on the coordination between socio-economic cost, resource conservation, and environmental protection. Second, based on “2012 China Sustainable Development Report” (2012) and other studies of the sustainable development of China, we can find out that the challenges of China's sustainable development are also mainly from three aspects, including high resource consumption, high social- economic costs, and serious environmental pollution. Moreover, based on literature review, we find the input factors mainly comprise natural resource, socioeconomic, and environmental factors, which have been explained together with necessary references in detail in the section of conceptual framework.

The above explanations have been added in the paper. Please check the revised version in Line 155-163.

Specific Comments 2

There are still many typos like ln121, "stduy."

Response: Thanks for your suggestion. We are very sorry for the typos. We have modified the typo "stduy" in L119. And we conducted a full English proof-reading, to improve the language of the paper. And we fixed some typos in the manuscript.

Specific Comments 3

A space is missing before the brackets for the reference. For example, in ln128, grown[23] should be grown [23].

Response: Thanks for your careful review. We have checked all the literature numbers and added spaces before the brackets.

Specific Comments 4

ln665. SFA is perhaps the stochastic frontier analysis but this is not explained anywhere in the paper. Again the authors avoid using abbreviations without providing the full word somewhere in the text.

Response: Thanks for the suggestion. We have added the full name of the SFA method in the paper. Please check the revised version in Line 594.

Specific Comments 5

A minor English proof editing is necessary before the final submission.

Response: Thanks for your suggestions. We do conduct a full English proof-reading, to improve the language of the paper. And we fixed some typos in the manuscript.